# Real-Time AI-Assisted Push-Broom Hyperspectral System for Precision Agriculture

**DOI:** 10.3390/s24020344

**Published:** 2024-01-06

**Authors:** Igor Neri, Silvia Caponi, Francesco Bonacci, Giacomo Clementi, Francesco Cottone, Luca Gammaitoni, Simone Figorilli, Luciano Ortenzi, Simone Aisa, Federico Pallottino, Maurizio Mattarelli

**Affiliations:** 1Department of Physics and Geology, University of Perugia, Via A. Pascoli, 06123 Perugia, Italy; 2Materials Foundry (IOM-CNR), National Research Council, c/o Department of Physics and Geology, Via A. Pascoli, 06123 Perugia, Italy; 3Consiglio per la Ricerca in Agricoltura e l’Analisi Dell’Economia Agraria (CREA)—Centro di Ricerca Ingegneria e Trasformazioni Agroalimentari, Via della Pascolare 16, Monterotondo, 00015 Rome, Italy; 4Department of Agriculture and Forest Sciences (DAFNE), Tuscia University, Via S. Camillo De Lellis, Via Angelo Maria Ricci, 35a-02100 Rieti, 01100 Viterbo, Italy

**Keywords:** artificial intelligence, crop monitoring, hyperspectral imaging, push-broom spectrometer, precision agriculture

## Abstract

In the ever-evolving landscape of modern agriculture, the integration of advanced technologies has become indispensable for optimizing crop management and ensuring sustainable food production. This paper presents the development and implementation of a real-time AI-assisted push-broom hyperspectral system for plant identification. The push-broom hyperspectral technique, coupled with artificial intelligence, offers unprecedented detail and accuracy in crop monitoring. This paper details the design and construction of the spectrometer, including optical assembly and system integration. The real-time acquisition and classification system, utilizing an embedded computing solution, is also described. The calibration and resolution analysis demonstrates the accuracy of the system in capturing spectral data. As a test, the system was applied to the classification of plant leaves. The AI algorithm based on neural networks allows for the continuous analysis of hyperspectral data relative up to 720 ground positions at 50 fps.

## 1. Introduction

In the ever-evolving landscape of modern agriculture, the integration of advanced technologies has become indispensable for optimizing crop management and ensuring sustainable food production. One such cutting-edge technology that has gained prominence in precision agriculture is the push-broom hyperspectral technique [1,2,3]. Unlike traditional multispectral imaging systems, push-broom hyperspectral sensors offer a continuous and high-resolution spectral data-set. This characteristic allows for a more comprehensive analysis of the optical reflectance spectrum, enabling the precise identification and characterization of various materials, including plants and their health indicators [4,5,6]. This innovative approach to remote sensing, when coupled with the power of artificial intelligence (AI), holds great promise for enhancing our ability to monitor and manage crops with unprecedented detail and accuracy [7,8,9,10,11].

Based on the huge spectral resolution employed, this analysis can provide additional sensitivity for agricultural applications. Pereira et al. demonstrated the feasibility to detect early bacterial disease onset on tomato using hyperspectral point measurement [12]. Gold et al. proved the efficacy of hyperspectral measurements for the detection of presymptomatic late blight and early blight in potato [13]. Multispectral devices for field applications are nowadays commercially sold all over the world [14,15,16]. However, the hyperspectral approach is often used for research purposes, aerial/UAV mapping, or proximal mapping, while not for real-time applications. This is mainly due to the heavy computational power required to elaborate the huge amount of data produced by imaging sensors. In this respect, AI is going to have a huge technological impact on the agrifood and forestry sectors, due to its ability to extract synthetic information from a large amount of collected data [17]. As a result, it allows us to face old problems in a new and effective perspective [18,19,20]. In particular, some authors have used AI algorithms for the extraction of synthetic indices [21,22] and to perform early detection on biotic and abiotic stress in rocket leaf [23]. Moreover, AI can push hyperspectral analysis towards real-time applications when combined with open source and commercial spectrometers [24,25,26].

The present work aims to develop and implement a real-time AI-assisted push-broom hyperspectral system for real-time applications to be used on a on a UGV (unmanned ground vehicle) featuring an innovative ultra-wideband autonomous navigation system. The system is designed to produce a real-time output signal used by an actuator to operate an action such as the site-specific application of a control treatment in a greenhouse or open field. We start from a meticulous analysis of the resolution in both the spectral and spatial domains, comparing the data generated by this hyperspectral system with that of a conventional commercial point spectrometer. As a test-bench, we then use hyperspectral data collected from lettuce and arugula leaves to train a neural network able to classify the plant species based on its spectral characteristics. The developed system showed the ability to classify the plant species used for the test with high accuracy (i.e., 0.996) at a high frame rate.

## 2. Materials and Methods

### 2.1. Push-Broom Spectrometer Design

#### 2.1.1. Optical Assembly

In a line spectrograph, the main elements are (1) the collection optics, which determine the investigated region, (2) the dispersing element responsible for the separation of the spectral features, and (3) the recording device, with suitable speed and sensitivity. The features of the different elements have to be chosen in order to satisfy the application requirements. Specifically, the real-world application that informed the design of the system was monitoring crops cultivated in 1 m wide strips, with the optical device positioned on an unmanned ground vehicle (UGV) moving longitudinally along the strip. Ensuring a spatial resolution of at least 1 cm^2^ is imperative to enable the discernment of weeds or diseases.

In the following, we detail the main elements of the push-broom spectrometer, as sketched in Figure 1:(1)A wide-angle (81°) objective, TTartisan APS–C 17 mm F1.4 [27] (TTArtisan Tech Co., Limited, Shenzhen, China), collecting lens L1, focuses the incoming light on a 20 mm long and 200 μm wide slit, 3D-printed in black PLA. Considering the objective focal length (f=26 mm, considering the crop factor), at a distance of 1.2 m, the slit selects on a soil a line which is about 1 cm wide and 1 m long. It should be noted that in such conditions, the depth of the field is about 25 cm, sufficient to accommodate the different heights of plants. Next, the slit and the collimating lens L2, f=75 mm [28] (Thorlabs Inc., Newton, NJ, USA) focus on the slit and collimate the light toward the prism.(2)An F2 equilateral prism [29] (Thorlabs Inc., Newton, NJ, USA) was chosen for dispersing the collected light. For this application, the prism presents an advantageous alternative to grating by offering simplicity and robustness, important features for a setup that can be mounted on a ground vehicle moving on rough terrain, also avoiding the complexities associated with higher diffraction orders. The light is dispersed by the prism in a direction perpendicular to the slit length so that, after the prism, the light rays’ vertical angle with the optical axis depends on the position with regard to the soil and the horizontal angle on the wavelength (mainly).(3)The re-imaging lens L3, f=25 mm [30] (Edmund Optics Inc., Barrington, NJ, USA), focuses the parallel light rays on the detector so that the horizontal coordinate of the sensor depends on the wavelength while the vertical component depends on its position with regard to the soil. The two lenses, L2 and L3, are in a telescopic configuration with a magnification factor equal to the ratio of the focal distances (M=1/3). The sensor is the monochrome camera Allied Vision Alvium 1800 U-040m [31] (Allied Vision, Stadtroda, Germany). It satisfies the requirements of a continuous acquisition and real-time analysis (max. frame rate at full resolution, 495 fps), together with the needed spectral and spatial resolution (728 × 544 px). In fact, at 50 fps, considering a UGV speed of 5 km/h (i.e., ∼14 cm/s), each snapshot differs by less than 3 mm, enough to measure the changes in different leaves. Moreover, the number of pixels allows for a nominal spatial resolution of 0.16 cm/px, with an average nominal spectral resolution in the sensitivity region of the detector (300–1000 nm) lower than 2 nm/px.

The final design was refined using 3DOptix (https://www.3doptix.com/, accessed on 19 June 2023) [32] (3DOptix, Rehovot, Israel). Optical simulation results are reported in Figure 2.

To investigate the performances of the spectrometer, we simulated three light sources laying in a line along the scanned dimension. The light is focused by the collecting lens on the slit and then collimated and dispersed by the prism in each color component. The re-imaging lens focuses the light on the detector. The output of the detector is pictured in the top-left corner of Figure 2. Each original point light source corresponds to a line on the detector, while the light is dispersed on its frequency component. We can observe that the hyperspectral image is subject to a distortion, where the wavelength positions are not constant for all the scanned dimensions. This distortion depends on the arrival angles on the detector along the scanned dimension and can be easily corrected by remapping the hyperspectral image.

#### 2.1.2. 3D Printing and Machining

The system support and the enclosure were designed using the CAD software SolidWorks 2017, as shown in Figure 3. The position of every optical element such as the objective, slit, prism, lenses, and camera is fixed into a base following the optical path, as shown in Figure 3.

The base is attached over a larger base that is connected to an enclosure designed to be resistant to humidity and water splashes. Figure 4 (right) and (left) show the photos of two prototypes of the spectrometer, one 3D-printed and the other machined. The 3D-printed version was realized in tough PLA, printed with an Ultimaker 3. The second prototype was realized in aluminum by CNC machining. In fact, the 3D-printed version is more convenient to fabricate and it is lightweight; however, at a high frame rate, it suffers from overheating problems and misalignment due to the thermal deformation of PLA. Indeed, heat production from the camera sensor is proportional to the camera frame rate; thus, for the high-throughput application fo thermal-resistant polymers, the 3D-printable or aluminum versions are preferable. In the present case, we chose to use an aluminum machined prototype, which is more heavy but also more robust and better suited for a UGV moving on rough terrain.

### 2.2. Real-Time Acquisition and Classification System

#### 2.2.1. Acquisition System

The custom-built real-time acquisition and classification system used for this project was developed with the single-board computer Raspberry Pi 4 Model B [33]. The Raspberry Pi module encompasses features such as a Micro SD port for external storage access, Bluetooth, wireless LAN, USB ports, and GPIO pins for external communication. The detector was connected to the Raspberry module through a USB 3.0 port in order to support its maximum frame rate of 495 frames per second, with eight bits per pixel (Mono8), at transmission speeds higher than 250 MByte/s. The system was installed with the Linux-based Raspberry Pi OS (Bullseye) distribution. To access data from the camera, we used the Vimba SDK v6.0 for Linux ARMv8 64-bit [34] (Allied Vision, Stadtroda, Germany), which provides APIs for C, C++, .NET, and Python. Each hyperspectral image acquired was converted in a NumPy matrix and then remapped with OpenCV [35] to correct the distortions introduced by the dispersive element of the optical train. The wireless LAN module of the Raspberry Pi was used to remotely communicate the classification results using Apache Kafka [36] to provide unified, high-throughput, low-latency, real-time data feeds.

#### 2.2.2. Data-Set

For training and classification purposes, we set up a data-set of reflectance spectra from two different plant species: lettuce and arugula. The reflectance spectra were recorded using the designed spectrometer. We placed several leaves from the plant along the scanned direction. The plant leaves were placed on a black surface with a white reference panel. The images were captured in natural, environmental light conditions. Reflectance was automatically calculated for each point along the scanned dimension, dividing the spectrum acquired on the leaves point by point by that acquired on the white panel, which was considered to have a constant reflectance equal to 1. This allowed us to take into account both the variability of the natural illuminator and the spectral responsivity of the whole system. Due to the absence of sharp peaks on the reflectance spectra (see Section 3.2) and to reduce the noise in the spectrum, the number of spectral features was reduced to 50, each feature comprising a range of about 10 nm in wavelength. Moving the spectrometer along the motion direction, we acquired multiple spectra for a total of 296,862 spectra, 224,867 from lettuce and 71,995 from arugula. For the training procedure, the data-set was split into random train and test subsets in the proportion 0.7 and 0.3, respectively. Samples in the data-set are reported in Table 1.

#### 2.2.3. Training and Classification

Training and classification were performed with scripts in Python, version 3.9, using the scikit-learn library [37], version 1.2.2. Hyperspectral data from the detector were split into single spectra, each one relative to a different space region observed. The classification problem can be solved with different techniques, as a decision tree or a neural network. A preliminary test using decision trees showed a good classification accuracy, but the generated structure resulted over-complex, with the risk of loosing generality. The model selected for the classification tests is the multilayer perceptron classifier (MLPClassifier). We tested several architectures, with different numbers of hidden layers and nodes per layer. With the aim of maintaining good accuracy in the classification and low computing complexity, we chose an architecture with one hidden layer consisting of 25 nodes. The neural network architecture schematic of the used classifier is reported in Figure 5. The output *y* of each node is given by
(1)$$y=g\left(\sum_{i=0}^{d}w_{i}x_{i}+bias\right)$$
where $w_{i}$ is the weight between layer *i* and layer i+1, $x_{i}$ is the *ith* input of the node, and bias is the values added to layer i+1. The function g(·) is a nonlinear activation function. The rectified linear unit function (relu) was used as an activation function for each node in the hidden layer, while the logistic sigmoid function (logistic) was used for the output layer. The weight optimization was performed with the stochastic gradient-based optimizer (adam) [38], with a maximum number of iterations equal to 500. The accuracy of the classification was validated using a computing confusion matrix, along with the main classification metrics (precision, recall, and F-measure). Once trained and tested, the classifier was validated, implementing a classification function for the real-time hyperspectral data in a scenario with mixed plant species.

## 3. Results

### 3.1. Calibration and Resolution Analysis

The wavelength-to-pixel calibration in a spectrometer is a crucial process that establishes a precise correspondence between the detected wavelengths of light and the corresponding pixels in the sensor of the spectrometer. This calibration ensures accurate and reliable spectral measurements by aligning the pixel positions with specific wavelengths of incoming light. Typically achieved through the use of known spectral lines or standard calibration sources, this calibration process compensates for any variations or distortions in the optical system. We performed the calibration procedure using the emission lines of a fluorescent lamp, three laser diodes (red, green, and blue), and an IR diode as the known sources. All these sources were recorded both with the commercial spectrometer and with the developed spectrometer, extracting the peak position in wavelength and pixel, respectively. The obtained experimental wavelength-to-pixel relationship is presented in Figure 6.

The wavelength-to-pixel relationship depends on the exit angle of the prism element, $\theta_{out}$, which is a function of the wavelength-dependent refractive index, n(λ), as
(2)$$\theta_{out}=\theta_{0}+\arcsin\left(n\left(\lambda \right)\sin\left(\alpha -\arcsin\left(\frac{1}{n(\lambda )}\sin\theta_{0}\right)\right)\right).$$

Considering a transparent medium, the refractive index and wavelength relationship can be described empirically using Sellmeier’s equation:(3)$$n^{2}\left(\lambda \right)=1+\sum_{i}\frac{B_{i}\lambda^{2}}{\lambda^{2}-C_{i}}$$
where $B_{i}$ and $C_{i}$ are experimentally determined Sellmeier’s coefficients. Simplifying Sellmeier’s equation to a one-term form, we obtain
(4)$$n^{2}\left(\lambda \right)=A+\frac{B\lambda^{2}}{\lambda^{2}-C}.$$

Linearizing Equation (Equation 2) around the refractive index of yellow and combining that with Equation (Equation 4), we obtain the wavelength-to-pixel relation:(5)$$pixel={\left(A^{\prime }+\frac{B^{\prime }\lambda^{2}}{\lambda^{2}-C}\right)}^{1/2}+D.$$

The obtained relation, fitted with experimental data, is plotted as a solid line in Figure 6, showing a good agreement between the data and the model.

In the following, we outline the procedures employed to validate the calibration of the system along both the frequency and spatial axes, also presenting the achieved spectral and spatial resolutions. The frequency response of the spectrometer was tested, measuring the spectrum of a fluorescent lamp, selected for its distinct spectral features.

The spectrum of the emitted light was recorded utilizing the presented system; the corresponding data are depicted in Figure 7 as a red line. As a comparative benchmark, the same measurement was performed using a commercial spectrometer, the Avantes AvaSpec-ULS2048-USB2-UA-50 (Avantes B.V., Apeldoorn, The Netherlands), working in the UV-VIS-NIR range. The resulting data are shown in the same figure, as a black line. The excellent agreement in the peak positions of the two spectra substantiates the accuracy of the calibration in the wavelength scale of the custom-made system. Notably, the discernible differences in the peak widths reflect the different frequency resolutions of the two spectroscopic systems. To estimate the frequency resolution, Δλ, of our setup, we used the width of the sharp peak in the emission spectrum located at about 540 nm. The so obtained value, less than 20 nm, can be considered a good estimation of Δλ. This value perfectly meets the requirements for our specific application, ensuring the system’s appropriateness for the designated scientific objectives, as the reflectance spectra of the leaves exhibit broad features rather than sharp peaks.

The contrast and the spatial resolution of the custom-made system were determined by analyzing the images of line patterns and performing a quantitative analysis on the obtained measurements. The black-and-white line pattern chosen for this purpose is reported in Figure 8a. It presents successive lines, varying their spatial frequencies in order to cover a range of spatial resolutions. The identification of the single lines in the captured image can be estimated by the plot of the intensity profile measured using the instrument and reported in Figure 8b. While thicker and more widely spaced lines are well resolved, as they converge, diminishing their thickness, as and the system’s ability to distinguish them separately decreases. Particularly, when the size of the single lines falls below 0.06 cm, as in the last line pattern, a single intensity peak is measured. The contrast parameter defines the system’s ability to distinguish separate lines of a given pattern. It is defined as
(6)$$C=\frac{I_{\max}-I_{\min}}{I_{\max}+I_{\min}}$$
and it was evaluated for the different groups of lines. The obtained values are reported in Figure 8c. The resulting spatial resolution estimated at C=0.5 corresponds to ∼0.5 cm.

Moreover, the evaluation of the linearity within the spatial sampling involves the examination of the distances in pixels between two consecutive maxima in the measured intensity in Figure 8b. The corresponding data are presented in Figure 8d alongside the linear fit. The noteworthy agreement underscores that the optical components selected for the system construction demonstrate an absence of measurable optical aberrations. It is worth noting that, in accordance with the theoretical expectations based on the arrangement of the chosen optical components, a single pixel of the camera corresponds to 0.1 mm in real space at 1 m.

### 3.2. Plant Classification Training, Tests, and Validation

As a stringent test for our system, we selected leaves from lettuce and arugula. In fact, while these leaves are easily recognizable from their shape, discriminating them only by their reflectance spectrum is a challenging task.

In Figure 9, we present the typical normalized reflectance spectra of lettuce (left) and arugula (right), captured at various angles and on different parts of the leaf. Each spectrum is depicted with a different color in order to make them distinguishable. As expected, the majority of the visible-range reflectance takes place in the green region for both species, exhibiting similar characteristics, with the primary distinction lying in the amplitude of the peak value within this green range. Nevertheless, relying solely on this feature proves inadequate for species discrimination due to the significant variation and overlap in this particular value between the two species. To address such complexities, we employed an artificial intelligence system based on neural networks.

The neural network architecture was trained on the data-set presented earlier. The data-set was split into random train and test subsets in the proportion 0.7 and 0.3, respectively, for evaluating the training procedure. Figure 10 reports the confusion matrix from the test set, which shows an accuracy of 0.996.

In Table 2, precision, recall, and F-measure are reported for each class. From the recall column, since we are dealing with a binary classifier, we can compute the sensitivity and specificity, which are both close to one.

The validation of the trained model was performed on a real-time data consisting of mixed leaves of lettuce and arugula, as pictured in Figure 11 (top image), where the region investigated using the push-broom spectrometer is highlighted; the scanned portion of the image is represented in color space. An example of the output classification is reported in bottom part of Figure 11, where different colors represent the classification outcome (green for lettuce, red for arugula). Before classification, we applied a filter to the spectra to skip those relative to the ground, thus granting them low integrated diffuse reflectance. These regions are reported in the classification output in gray. As evident from the validation results, in some cases, the leaves are misclassified. This is probably due to the proximity of the leaves of different species, whose spectra are partially reflected by the whiter part of the leaf.

## 4. Discussion

As we have seen in the spectra used for the analysis, the reflectance spectrum of plant leaves exhibits significant variability, even within a confined spatial area. This variation is influenced by various factors. Some of these factors depend on the intrinsic features of the plant, including the pigments of the leaf, cell structure, water content, and biochemical composition [39,40,41]. While chlorophyll, the primary pigment responsible for photosynthesis, strongly absorbs light in the blue and red regions of the spectrum, giving leaves their characteristic green color, other compounds, such as carotenoids and anthocyanins, contribute to the overall reflectance pattern [42,43]. Moreover, the presence of pathogens can significantly alter the reflectance spectrum of plants due to the physiological and biochemical changes induced by the infection. Pathogens, such as bacteria, fungi, and viruses, can cause structural alterations and trigger biochemical responses in plant tissues [12,13]. These changes can affect the absorption and reflection of light. For instance, pathogens may cause disruptions in chlorophyll content, affecting its absorption peaks and resulting in an overall shift in reflectance patterns. Additionally, pathogen-induced stress can lead to changes in water content and cell structure and to the accumulation of secondary metabolites, all of which contribute to alterations in the reflectance spectrum. Other features of the collected spectrum depend on the “experimental” conditions. The reflectance spectrum of a leaf can change based on the observation point, which includes both the angle and position of the incident light and the viewing angle [44,45,46]. When the observation point changes, the interaction of light with the surface of the leaf structure and pigments can lead to variations in the reflectance spectrum. As the angle of incidence increases or decreases, the amount of light that is absorbed, transmitted, or reflected by the leaf changes. This is influenced by factors such as the orientation of the leaf, the surface roughness, and the arrangement of the cells and pigments. To capture a comprehensive view of the properties of place reflectance, is it usually necessary to use various angles and positions when conducting spectral analysis. Moreover, the natural spectral source is not constant; it can change because of the time of the day or the atmospheric conditions. In a real scenario, as in on field measurement of crop status, the excellent results on classification obtained in the controlled environment of the laboratory cannot be expected in general. In this complex scenario, the classification strategies often have to be optimized on the spot. These problems can be sometimes be overcome by increasing the complexity of the classification model. The classifier used in the present work is based on neural networks, which indeed improve the model’s performance and are less prone to overfitting [47]. Generalization is also aided by the data acquisition process, which samples a large variety of illumination conditions. VIS-NIR spectra are very broad, and it is more important to recognize the spectral signature of this phenomenon than for single sharp peaks.As a result, the spectral resolution can be acceptably reduced in favor of a short acquisition time. In this way, many realizations can be collected, and the influence of external conditions on the measure can be reduced. Also, the noise in the spectrum is reduced, averaging out in the adjacent wavelength region. Thus, the present experimental setup results as an intermediate between a point-based setup, characterized by high spectral resolution and being very site specific, and that of a hyperspectral camera, very informative but very slow. The system presented here also allows for a continuous calibration of the actual reflectance spectrum features taken into account for real-time analysis. In fact, on the one hand, the training of the classification system can be carried out in few minutes, also using the limited capabilities of the embedded Raspberry Pi; on the other hand, all the acquired spectra can be corrected considering the instantaneous illumination condition, as the reference spectrum can be always acquired once a white reference panel, traveling with the UGV, is placed in the field of vision of the camera.

In Table 3, we report the main specifications of the developed hyperspectral system. Note that the specificity of the application and the versatility in the analysis allow for relaxing the requirements in terms of the spectral resolution of the setup. In fact, leaf reflectance spectra usually present smooth features. Conversely, it is crucial to minimize the acquisition and elaboration time. This parameter is of pivotal importance in order to enhance the velocity of the UGV for any future applications. This can be achieved thanks to the small number of spectral features observed, reducing the computational effort required for the analysis in order to meet the requirements for the real-time application of the system.

## 5. Conclusions

In this work, we presented the development of a real-time system for plant spectrum classification from hyperspectral data. The developed system encompasses a custom-designed push-broom spectrometer where the spectral resolution is traded in favor of light collection to obtain a high frame rate. The data are processed and classified by a neural network model in an embedded system. The fast acquisition time, in conjunction with the efficiency of the neural network architecture, permits the real-time analysis of the reflectance spectra. As test example of its application, the system was trained to classify leaves from different plant species (i.e., lettuce and arugula), obtaining the classification of more than 35,000 spectra per second with an accuracy of 0.996. The same system can potentially be trained and used to detect weeds or diseases in a greenhouse or open-field plantation. Future studies will implement the sensor on a UGV featuring an innovative ultra-wideband autonomous navigation system for automated greenhouse treatment.

## Figures and Tables

**Figure 1 sensors-24-00344-f001:**
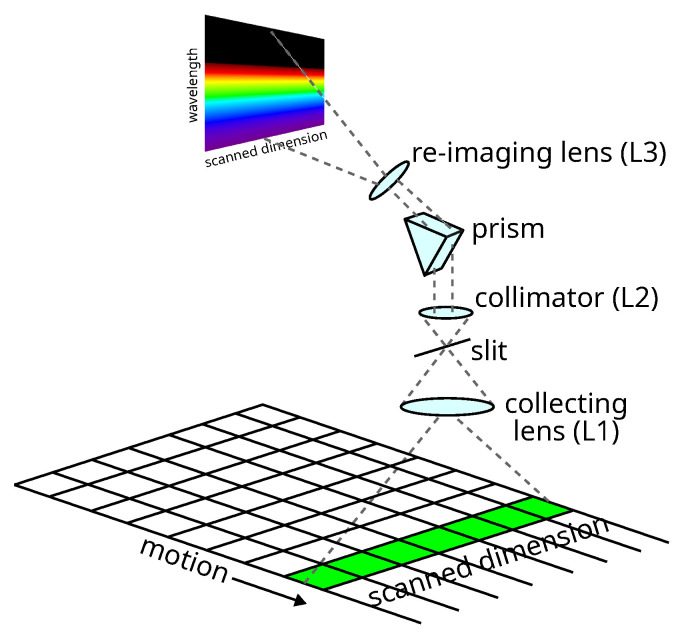
Push-broom spectrometer working principle. Collecting lens focuses the target on a slit and selects only a portion of the target. The light is then dispersed by a prism and refocused on the imaging sensor.

**Figure 2 sensors-24-00344-f002:**
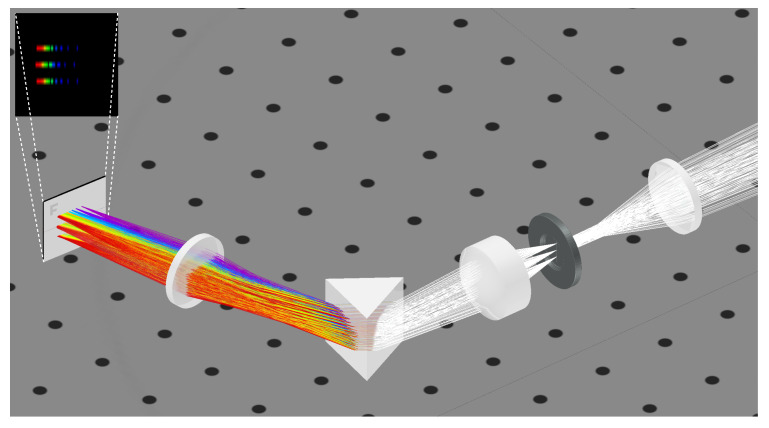
The 3D optical simulation of the push-broom spectrometer. Simulation was performed with 3DOptix software, v2.0. For representative purposes, the incoming white light starts from three points. The light is focused on the vertical slit and then dispersed by the prism and finally refocused on the sensor. Top-left inset shows three strips of dispersed light on the detector relative to the three starting points.

**Figure 3 sensors-24-00344-f003:**
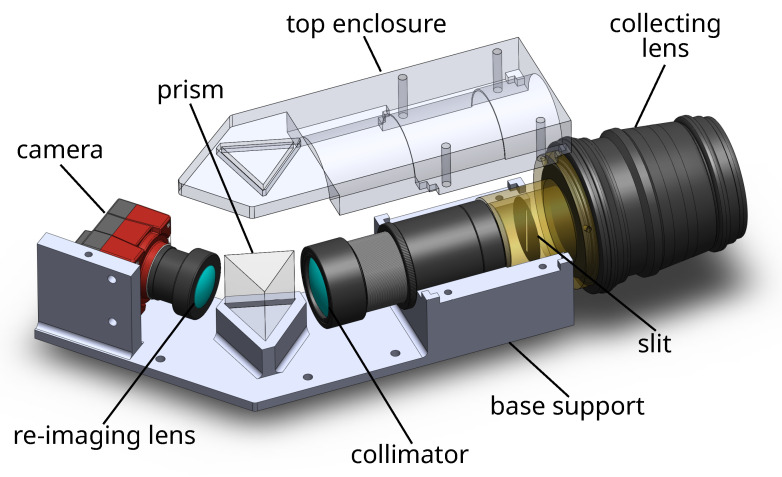
CAD drawing of the system with base support and optical elements.

**Figure 4 sensors-24-00344-f004:**
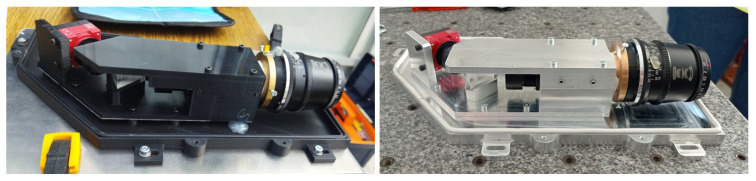
The 3D-printed (**left**) and aluminum CNC-machined (**right**) versions of the prototype.

**Figure 5 sensors-24-00344-f005:**
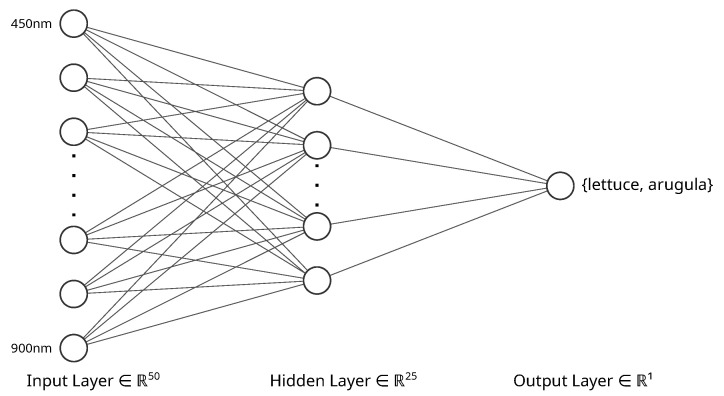
Neural network architecture schematic. The input features consist of the normalized reflectance at different wavelengths. A hidden layer with 25 node was weighed and combined with the spectral points. The binary output layer classified the spectra based on the results of the hidden layer.

**Figure 6 sensors-24-00344-f006:**
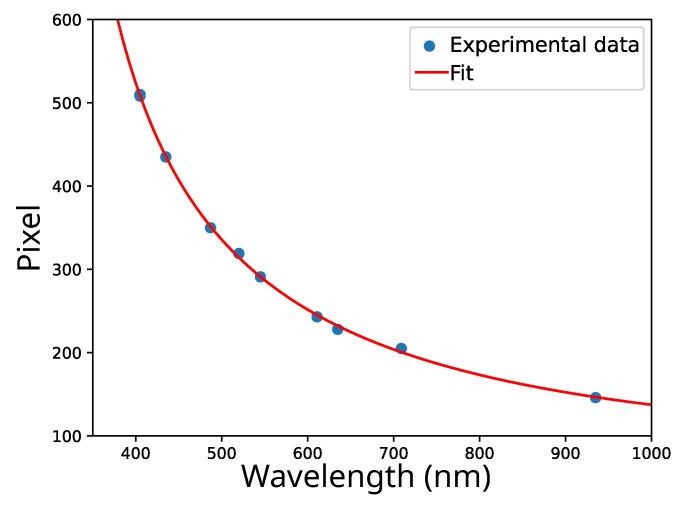
Wavelength-to-pixel relationship obtained from the position of the peaks of a fluorescent lamp spectrum, laser diodes, and IR diode. Dots represent experimental data and the solid line represents the fit according to Equation (Equation 5).

**Figure 7 sensors-24-00344-f007:**
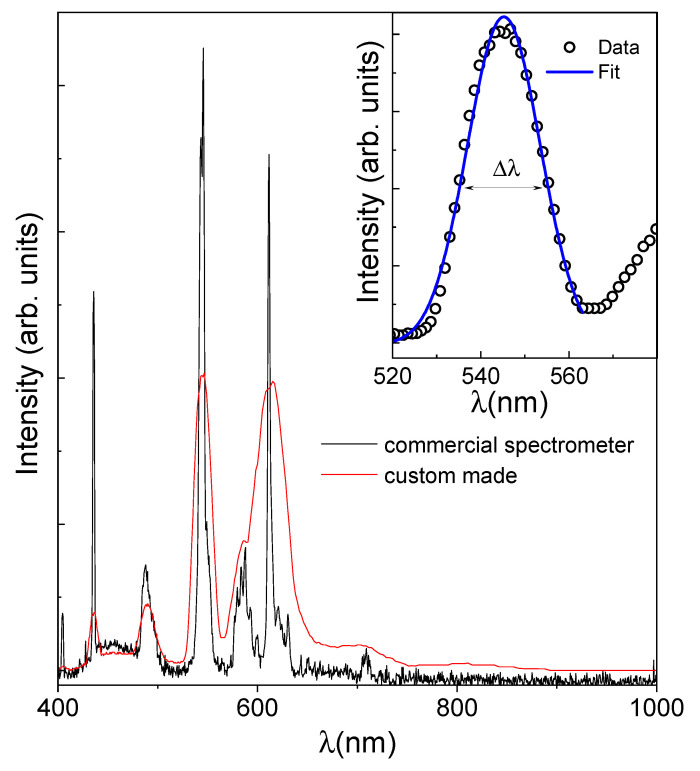
Comparison of the spectrum of a fluorescent lamp acquired using a commercial spectrometer (Avantes AvaSpec-ULS2048-USB2-UA-50) and the custom-made spectroscopic system. Inset: The spectral resolution of the system, δλ, can be estimated using the measured width of a very sharp peak. The estimation was conducted with an atomic emission line located at about 540 nm of the fluorescent lamp.

**Figure 8 sensors-24-00344-f008:**
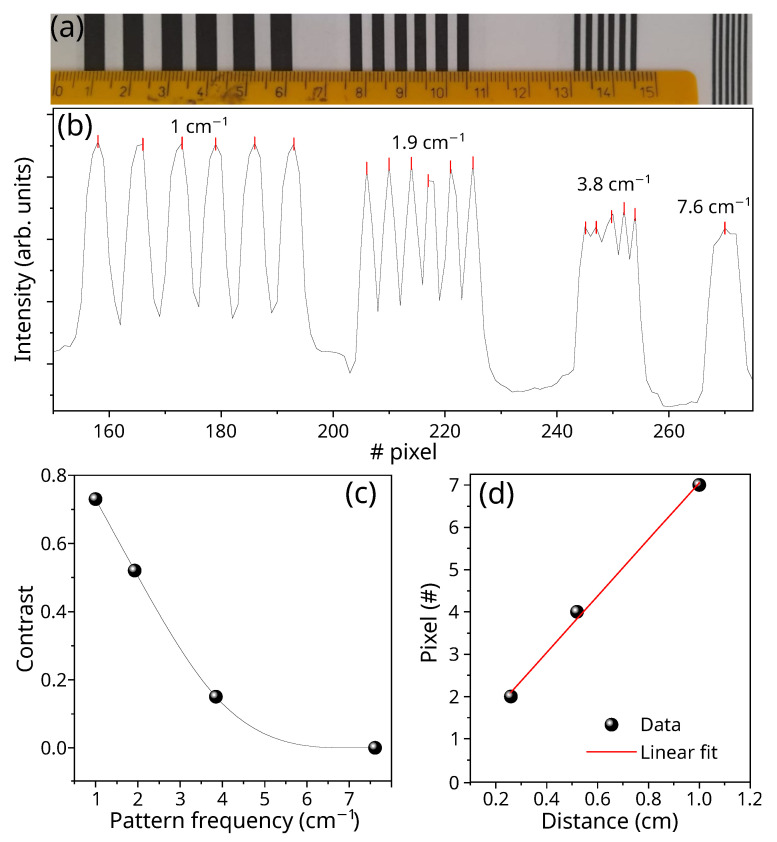
Spatial resolution of the spectrometer. (**a**) Line pattern used to calibrate the spatial response function and the contrast of the instrument. (**b**) Measured intensity as a function of the pixel position. (**c**) Contrast of the four-line patterns as a function of the frequency of the chosen pattern (dots); the line is a guide for the eyes. (**d**) Distance in pixels between two successive maxima as a function of the dimension of the pattern period (black dots) together with the linear fit (solid line).

**Figure 9 sensors-24-00344-f009:**
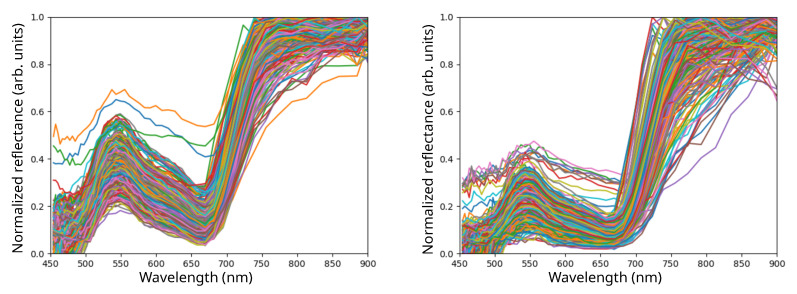
Normalized reflectance spectra acquired from leaves of lettuce (**left**) and arugula (**right**) at different angles and positions. Each set of spectra shows a large variance due to the part of the leaf and acquisition angle selected.

**Figure 10 sensors-24-00344-f010:**
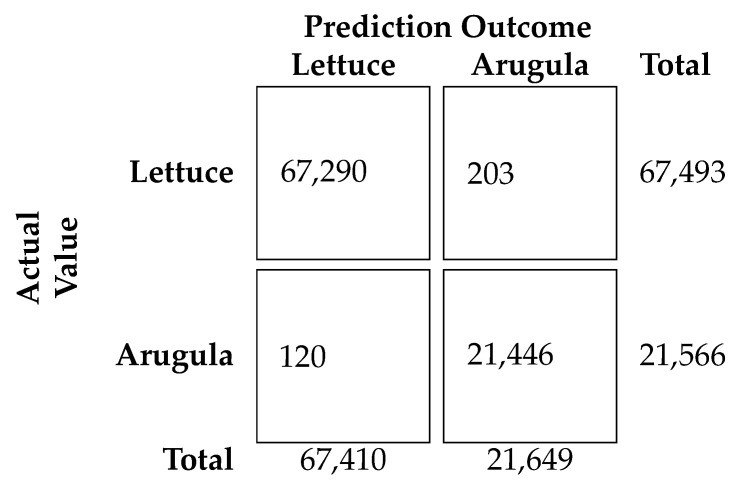
Confusion matrix for lettuce and arugula classification. The rows correspond to the true class, and the columns correspond to the predicted class. Diagonal and off-diagonal cells correspond to correctly and incorrectly classified observations, respectively.

**Figure 11 sensors-24-00344-f011:**
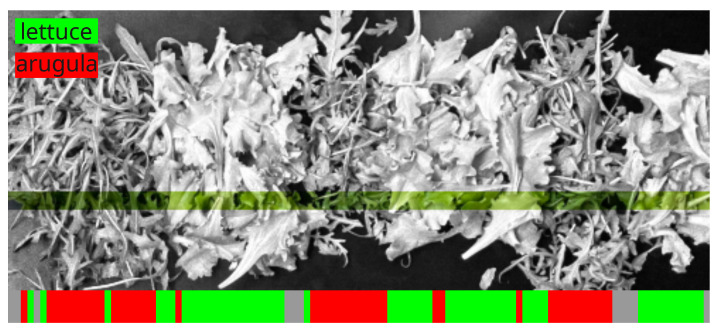
Classifier validation on mixed plant leaves. (**Top**) panel shows an image of the leaves where colored region represents the dimension scanned by the push-broom spectrometer. (**Bottom**) panel shows an example of classification (green for lettuce and red of arugula) where most of the points are classified correctly.

**Table 1 sensors-24-00344-t001:** Samples in the data-set. The data-set contains 296,862 spectra, 224,867 from lettuce and 71,995 from arugula. The data-set was split into random train and test subsets in the proportion 0.7 and 0.3, respectively.

Samples	Lettuce	Arugula	Total
Train	157,407	50,396	207,803
Test	67,460	21,599	89,059
Total	224,867	71,995	296,862

**Table 2 sensors-24-00344-t002:** Classification report (precision, recall, F-measure, and accuracy) for each class. From the recall column, we observe a sensitivity and specificity close to one.

	Precision	Recall	F-Measure
Lettuce	1.00	1.00	1.00
Arugula	0.99	0.99	0.99
Accuracy			0.996

**Table 3 sensors-24-00344-t003:** Summary of the technical data of the newly developed portable hyperspectral imaging device for remote sensing at the operating conditions. The table includes the spatial and frequency specifications, the principal physical parameters such as weight and dimensions, and the data treatment.

Relevant Working Parameters	
Wavelength operation range	300–1000 nm
Spectral resolution	<20 nm at 540 nm
Field of view at 1.2 m	Soil line 1 cm wide and 1 m long
Working distance	From 0.9 to 1.10 m
Spatial resolution	∼0.5 cm along the scanned dimension line
	∼1 cm along the motion direction
Acquisition time	Max. frame rate at full resolution, 495 fps
Classification speed	∼35,000 spectra @ 50 fps
Data transfer	Wireless connectivity
Weight	∼1.5 kg
Dimensions	30 cm × 20 cm × 10 cm (L × W × H)

## Data Availability

Data are contained within the article.

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
