# Peer review of "Real-Time AI-Assisted Push-Broom Hyperspectral System for Precision Agriculture"

_sensors, 2024, doi:10.3390/s24020344_

Round 1

Reviewer 1 Report

Comments and Suggestions for Authors

The paper provides an in-depth exploration of a push broom spectrometer's design, a real-time acquisition system, and an AI-based classification approach for plant leaves. This method achieves continuous hyperspectral data analysis for 720 ground positions at 50 frames per second (fps).

1. Enhance the 'Materials and Methods' section by incorporating additional specifics regarding the push broom spectrometer's design.

2. Clarify the preprocessing steps and consider presenting dataset details in a table for a more concise overview.

3. While the paper currently employs only one classifier (MLP), it is advisable to extend the analysis by comparing its performance with other deep learning algorithms, such as CNN.

4. Ensure to include accuracy metrics in the "training and classification test" section to provide a comprehensive evaluation of the model's performance.

These adjustments will contribute to a more comprehensive and detailed presentation of the research findings.

Reviewer 2 Report

Comments and Suggestions for Authors

This manuscript (sensors-2776016) introduces a real-time AI-assisted push broom hyperspectral system for plant identification and disease detection in agriculture, detailing its design, construction, and application. This approach emphasizes the system's ability to accurately capture spectral data and analyse up to 720 ground positions at 100 fps using neural networks.

The manuscript is intriguing, but its writing is not. The introduction needs improvement; it contains only one paragraph with a few sentences on the topic. The authors should supplement the information on hyperspectroradiometry, historical context, and its uses and applications. Then, they should discuss AI, followed by precision agriculture and its applications. It is also important to clarify the hypotheses and objectives of the work.

Moreover, the discussion and conclusion sections are inadequate, completely diverging from what is expected in a scientific manuscript. There is no connection to the relevant literature. What are the perspectives? What were the significant advancements? How does this new method compare to or improve upon what we currently use? For example, why are the curves very noisy? What other evaluations were conducted for validation? What was the sample size used?

In the results section, there's a mix of discussion. Keywords should be in alphabetical order, and the figure captions need a better description of all the elements present, so that viewing them does not require going back to the manuscript text.

The combined discussion and conclusion section is not suitable. They need to carefully and scientifically describe the results based on literature. References are outdated and need to be improved.

Comments on the Quality of English Language

The English requires moderate corrections in grammar, spelling, and verbosity.

Reviewer 3 Report

Comments and Suggestions for Authors

The topic of the presented research is actual and perspective. Thanks to the rapid development of information technology and robotics, it is becoming possible to use high-quality hyperspectral imaging and monitor plant health in real time.

The authors presented the article consistently, they discussed the main issues, but there are some recommendations:

1. It is necessary to clearly highlight the gap and key ideas of the study (or theses) at the end of the introduction.

2. Despite the fact that the authors provided references where programs or component models are mentioned, it is better to indicate the manufacturer and country in brackets (lines 65, 83, 91, 104, 110, 121, etc.).

3. It is recommended to add at least a brief description of the UGV being used.

4. Both in the introduction and in the abstract, the authors stated that an AI system for detecting diseases has been developed and implemented, but no results on this direction are presented in the article itself.

5. The article mentions a comparison of the characteristics of a custom camera with a commercial point hyperspectral camera; it is recommended to indicate the camera model (commercial).

6. Did the researchers noisy the original dataset (were leaves other than lettuce and arugula added)?

7. It is recommended to describe the neural network architecture used in a little more detail.

Round 2

Reviewer 2 Report

Comments and Suggestions for Authors

The authors have significantly improved the manuscript. Additionally, they have included discussions and reorganized relevant sentences that enhance readability. I believe that before publication, the authors should reorganize the captions of the tables to the top and complement them with as complete a description as possible, indicating everything that appears, both in the figure captions and the table captions, such as the number of samples, for example. Best regards.

Comments on the Quality of English Language

Moderate corrections in grammar and synthases.

Author Response

We would thank again the referee for the usefull comments. We have fixed the position of table captions and added further description.

The manuscript went under thoughtful English revision. We have corrected several grammar and syntax errors.

Reviewer 3 Report

Comments and Suggestions for Authors

The authors of the presented article took into account all previous comments and recommendations. The introduction has been rewritten, highlighting the key ideas of the study. Component and software manufacturers have been added. The authors added a new section with a description of the original dataset, as well as a clear schematic representation of the selected neural network architecture. The limitations of the study were formulated in the discussion section, and comments were corrected. I recommend the manuscript for publication.

I would like to take this opportunity to congratulate the journal editors and the authors: dear colleagues, Merry Christmas and Happy New Year!

Author Response

We would thank again the referee for the useful comments.